# Long-Term, Single-Centre Observation of Patients with Cardiac Implantable Electronic Devices

**DOI:** 10.3390/medicina57121357

**Published:** 2021-12-13

**Authors:** Roman Załuska, Anna Milewska, Anastasius Moumtzoglou, Marcin Grabowski, Wojciech Drygas

**Affiliations:** 1Department of Management and Logistics in Health Care, Medical University of Lodz, 90-131 Lodz, Poland; 2Department of Statistics and Medical Informatics, Medical University of Bialystok, 15-089 Bialystok, Poland; anna.milewska@umb.edu.pl; 3P. & A. Kyriakou Children’s Hospital, 11527 Athens, Greece; anastasius.moumtzoglou@gmail.com; 41-st Department of Cardiology, Medical University of Warsaw, 02-097 Warsaw, Poland; marcin.grabowski@wum.edu.pl; 5Department of Epidemiology, Cardiovascular Disease Prevention and Health Promotion, National Institute of Cardiology, 04-628 Warsaw, Poland; wdrygas@ikard.pl; 6Department of Social and Preventive Medicine, Medical University of Lodz, 90-419 Lodz, Poland

**Keywords:** atrioventricular conduction disturbances, chronic kidney disease, electrotherapy, hypothyroidism, prognosis

## Abstract

*Background and Objectives*: Electrotherapy is a valuable treatment method for patients with heart rhythm disturbances. There are very few observations of long-term patients treated with these techniques. There is a particular lack of this type of study conducted in Eastern European countries. The aim of this single-centre analysis was to evaluate the long-term survival (from 2010 to 2018) of patients treated with electrotherapy devices, taking into account clinical factors facilitating the prognosis of these patients. *Materials and Methods*: The patients (*N* = 2071) subsequently included in the study were subjected to the implementation or replacement of cardiac pacemakers. The medical records of all the patients were analysed. Data concerning death, made available by the State Systems Department of the Ministry of Administration and Digitization, were used. *Results*: The patients with VVI pacemakers had the worst prognosis after the replacement of the devices. Male patients had a worse prognosis, regardless of the kind of device implanted. Advanced atrioventricular conduction disturbances, chronic kidney disease, and hypothyroidism with reduced left ventricular ejection fraction were among the most significant coexisting diseases. *Conclusions*: The long-term prognosis of patients under different forms of electrotherapy remains poor. Despite the more straightforward technique, a single-chamber device (VVI/AAI) or generator replacement leads to the worst prognosis. The complexity of the clinical picture that stems from coexisting diseases and advanced age is of the utmost importance.

## 1. Introduction

Electrotherapy is a treatment for patients with heart rhythm disturbances and cardiac insufficiency. In recent years, there has been dynamic growth in technological solutions, programming methods, devices miniaturisation, and power supply optimisation.

Electrotherapy is commonly used all over the world. Implantable devices come from leading manufacturers who continuously modernise their technologies. Recommendations for device implantation in several clinical contexts are being updated in line with scientific development and featured in guidelines for scientific societies [1,2]. The most recent guidelines published in 2013 mentioned, among others, the clinical characteristics of patients who had single- and dual-chamber pacemakers implanted. The mean age of patients treated with this technique was 75–78 years, depending on the country. In Western European countries, the percentage of patients who had single-chamber pacemakers implanted ranged from 21.4% to 32.2% (dual-chamber: from 63.8% to 75.4%) [1].

Despite the long history of treating patients with these techniques, there are few long-term observations that include all types of devices used and evaluate the influence of coexisting diseases on the long-term prognosis of said patients. Moreover, there are no data available from Eastern European countries.

The present single-centre analysis was carried out based on data acquired from the Electrophysiology Laboratory of the Masovian Specialist Hospital in Ostroleka, where around 300 procedures of this type are performed annually. Over nine years, the analysis considered a large group of patients with a history of implantation of single- and double-chamber pacemaker devices, cardioverter defibrillators, and cardiac resynchronisation devices. Apart from the fact that these procedures were found to be prognostically beneficial, a significant influence of coexisting diseases on the patients’ lives became evident. The analysis carried out indicates the need for comprehensive treatment of that particular patient group. A constant improvement in medical treatment standards, economic conditions, and a change in patients’ lifestyles can positively impact the prognosis of patients who are being treated with these methods [3].

This study aimed to assess the long-term survival of patients treated with electrotherapy devices, considering the various clinical factors determining patient prognosis.

## 2. Materials and Methods

### 2.1. Study Population

Consecutive patients subjected to their first implantation procedure or device replacement over 2010–2018 were included in the analysis. All of the patients subjected to the guidelines had their medical records analysed. An electronic database and hospital treatment information cards were used, and information about survival was obtained from the State Systems Department of the Ministry of Administration and Digitization.

### 2.2. Statistical Analysis

The chi-square independence test was used in the statistical analysis to assess the relationship between the categorical variables. The normality of the distribution was verified by the Kolmogorov–Smirnov test, with Lilliefors modification, and the Shapiro–Wilk test. Comparing the quantitative variables with non-normal distribution in many groups, the nonparametric Kruskal–Wallis test by ranks (one-way ANOVA on ranks test) and a post-hoc multiple comparison test were implemented. Multivariable stepwise backward analysis of Cox’s proportional hazard model was implemented to analyse the accumulated independent variables’ influence on survival. Employing the determining variables of significant statistical impact on survival, a univariate analysis was executed, and a multivariate model was created. Those variables that violated the assumption were included in the model as stratifying variables.

The survival curves were determined using the Kaplan–Meier estimate. The level of significance was set at *p* < 0.05. Statistical calculations were made using Statistica 13.0 and Stata/IC 12.1 software.

## 3. Results

In total, 2071 patients were included in the study. The baseline characteristics of the studied patients, according to the device type division, are presented in Table 1. Table 2 shows the primary indications for device implantation. The impact of clinical factors on survival varied by group. In the SC-AAI/VVI population, chronic kidney disease and AV block III had the greatest impact, increasing the risk of death by 1.94- and 1.59-fold, respectively. In the DC-DDD group, an alternating bundle branch block increased the risk of death by almost 40-fold, while the heart failure III NYHA class increased the risk of death by 1.78-fold. Chronic kidney disease and diabetes had similar significance levels (an increase of approximately 1.6-fold). Atrial fibrillation with an AV block increased the risk of death by almost 5-fold in the ICD/CRT group. Hypothyroidism was also significant in these patients (an increase in risk of death of almost 3.5-fold). In the population with implanted pacing systems, males had a worse prognosis. An increase in age by one year had a similar prognostic significance in these groups and slightly less in the ICD/CRD patients. The results of the multivariate analysis are shown in Table 3.

Long-term survival varied according to the type of device implanted. In the SC-AAI/VVI group, approximately 39% of patients died during the follow-up period. The median survival was significantly lower than in the DC-DDD population, in which approximately 25% of patients died. In the ICD/CRT group, approximately 27% of patients died during the follow-up period. It was not possible to determine the median survival in this group, because the probability of survival was more than 50%. The median survival of patients who required pacing system replacement was significantly worse than for initial implantation, and similar regardless of device type. A flowchart of the study with long-term follow-up is presented in Figure 1.

The Kaplan–Meier survival curve, by device type and divided according to the first time of implantation and battery replacement, is presented in Figure 2. The likelihood of patient survival, comparing three types of implanted devices, differs from a statistical point of view (*p* < 0.001; Figure 2A). Patients treated with AAI and VVI pacing had a lower probability of survival than patients treated with DDD pacing, whose likelihood was the highest. It was typical of patients with the first implantation to have a higher survival probability than after a second one (Figure 2B). This difference was not statistically significant, but close to the statistical significance level of 0.05 (*p* = 0.055). There was no statistically significant difference in survival between patients treated with AAI/VVI pacing after the first and subsequent implantation (*p* = 0.353; Figure 2) and ICD/CRT (*p* = 0.532; Figure 2). However, in patients treated with DDD pacing, there was a statistically significant difference in the likelihood of survival during the first and subsequent implantations (*p* = 0.046; Figure 2). It was typical for patients with the first implantation to have a higher chance of survival.

## 4. Discussion

In recent years, a number of studies have been carried out worldwide, analysing the long-term survival of patients with implanted devices for electrotherapy. In most of them, the focus was on the type of devices implanted and the indications for treatment, as well as the influence of the stimulator on survival [4,5]. Only in some was the prognostic impact of coexisting diseases analysed, besides the technical aspects [6,7,8].

This work provides a comprehensive analysis of the impact on the survival of various factors, including the device implanted, type of surgery (first implantation or device replacement), age, sex, and a series of coexisting diseases. However, there are no data available from Eastern European countries.

Cardiovascular risk factors are commonly known. However, besides the typical factors of poor prognosis (e.g., heart failure NYHA class III), the multifactor analysis showed the importance of non-heart-related factors, such as hypothyroidism, distance from residence, length of hospitalisation, and kidney failure. In this study, the worst prognosis was for patients with single-chamber implants (VVI/AAI) and after device replacement. These treatments were performed in the oldest population, which could be a factor in survival.

Retrospective studies have shown that VVI stimulation, compared to other factors, may negatively impact prognosis and short-term observation. These results are consistent with those obtained from other analyses [6,8,9]. An increased incidence of clinically apparent heart failure was observed among patients with implanted VVI devices compared to DDD, which might explain the negative influence of the type of stimulation [10]. This may be changed by new trends in electrotherapy, such as stimulation of the middle part of the interventricular septum or, even better, His bundle stimulation or selective pacing of the left His bundle branch. Regardless of the type of stimulator and the treatment indication, the prognosis was worse in male patients. Similar results have been obtained in other analyses [7,11,12]. In the analysis, an increase in age of one year increased the risk by 6% in patients with stimulators and by 4% in the ICD/CRT device group. In other analyses, this risk was 5–9% for stimulators [7,11]. The present study showed the negative prognostic importance of advanced atrioventricular conduction disturbances (AV block III and atrial fibrillation with an AV block) in a younger population than in previously published work. Damage to the conductive system is usually the result of advanced cardiac disease. Both of these factors worsen survival. Stimulator implantation frees patients from symptoms and improves prognosis (especially in the AV block III degree). However, the risk of non-arrhythmia death remains increased [13,14]. Disorders of AV conduction were an independent risk factor for death in the population analysed by Shen et al. [15]. The study concerned patients in their eighth or ninth decade of life.

Chronic kidney disease had a significantly negative prognostic significance in the presented analysis in the population of patients with implantable pacing systems. There has been a lack of research on the relationship between chronic kidney disease and the survival of patients with implanted electrotherapy devices in younger age groups. This relationship was demonstrated in the present study. The risk of sudden cardiac death is related to the degree of GFR reduction. It increases by approximately 17% for each 10 mL/min/1.73 m^2^ GFR reduction; 15–20% of deaths in this group are caused by bradycardia, including an advanced AV block and asystole [16]. Fabian et al. [17] have shown that reducing the GFR reduces survival in the older population with implanted devices. Patients with an implanted CRT with a mild or moderate impairment of renal function benefitted from implantation in a similar way to those without chronic kidney disease. The device’s implantation had little effect on the deceleration of the deterioration of its function. Data on chronic kidney disease and ICD implantation in primary prevention are ambiguous. The benefits of treatment in secondary SCD prevention are well documented [18].

The present paper indicated an increased risk of death among patients with hypothyroidism and a low left ventricle ejection fraction (group ICD/CRT). According to a meta-analysis by Yang et al., the risk of death due to all causes in such a group is higher by 45% [19]. Among the population of patients with dilated cardiomyopathy, the risk of death is even higher. Similarly adverse are low T3 syndrome and subclinical hypothyroidism [20]. The cardiovascular system is significantly affected by thyroid hormones. Overt, as well as subclinical, hypothyroidism worsens the prognosis in chronic heart failure. Chronic cardiovascular diseases have an impact on the metabolism of thyroid hormones. The imbalance exacerbates the course of heart disease in a feedback mechanism; as left ventricular systolic dysfunction progresses, the reverse triiodothyronine (rT3) concentration increases, causing T3 and T4 levels to decrease [21]. In the advanced stages of heart failure, this phenomenon is the most intense. The deficiency of thyroid hormones causes structural and functional changes in the heart, leading to pulmonary hypertension and worsening the prognosis.

Moreover, it causes a decrease in preload and an increase in afterload. Due to these changes, the cardiac output is reduced. Secondary lipid disturbances intensify coronary atherosclerosis.

Among the analysed population, a 56% increase was seen in the risk of death in patients with diabetes in whom DDD pacing systems were implanted. As to concomitant diseases, an alternating bundle branch block, heart failure (NYHA class III), and chronic kidney disease appeared to dominate in this group. According to a multivariate analysis, these factors substantially worsen the prognosis. Chronic type 2 diabetes affects myocardium structural changes, accelerated development in atherosclerosis, the degeneration of cardiac conduction system, and autonomic neuropathy. The reasons listed above result in accelerated development of coronary disease, cardiac insufficiency, arrhythmia, and AV conduction. The most common cause of arrhythmia is atrial fibrillation. A higher frequency of sudden cardiac death during ventricular fibrillation has been indicated [22]. AV block III and sinus node function disorders occur more often in bradyarrhythmia than in the general population [22]. Patients’ age, the duration of diabetes and its poor control, BMI, and accompanying chronic kidney disease worsen the prognosis. Stimulator implantation necessity occurs more often than among the general population [23]. The risk of bradyarrhythmia increases during states of hypoglycaemia [24]. A surprising result of our analysis was the finding of positive prognostic significance of hypertension in the DDD-stimulated group. It is suggested that hypertension in the elderly may be an indicator of better cardiovascular health. However, this phenomenon requires further study.

Similar results have been observed in elderly patients (≥80 years). A study by Krzemień-Wolska et al. showed a 48% decrease in hypertensive patients’ mortality [8].

### Limitations

The present research has its limitations, i.e., its single-centre nature, no possibility of obtaining a broader clinical profile for all patients (e.g., pharmacotherapy data), lower ICD/CRT patient representation, and a short observation period. Another limitation of the present work is that laboratory parameters were not included, which might, to some extent, allow for predictions of the outcome. Nonetheless, all of the patients were assessed according to non-heart-related comorbidities, which form a certain equivalent to positive or negative laboratory results. Precise and detailed analysis of both coexisting illnesses and noncardiac illnesses and factors was conducted in this study. Thus, it was possible to indicate the most important prognostic factors based on the multivariable model.

## 5. Conclusions

The present study provided the results of a long-term observation of patients with implanted electrotherapy devices from a Polish centre. In the available literature, there is a lack of research dealing with matters similar to the ones pertaining to this study, with data from Eastern Europe and including, apart from indications for surgery, the type of implanted device, type of procedure (first implantation, device replacement), clinical data, and an array of coexisting diseases, which have an impact on survival.

Based on a multivariate model, it was shown that, besides some typical factors worsening the prognosis such as old age, advanced atrioventricular conduction disorders, heart failure (NYHA class III), and diabetes, noncardiac factors are of immense importance (chronic kidney disease, hypothyroidism, length of hospitalisation, and distance from the permanent residence). The worst prognosis was for patients who had single-chamber pacemakers implanted or had generators replaced, which is a testament to the complex clinical picture of the oldest segment of the population. New stimulation techniques of the right ventricle might positively impact the prognosis for these patients, particularly in younger age groups.

## Figures and Tables

**Figure 1 medicina-57-01357-f001:**
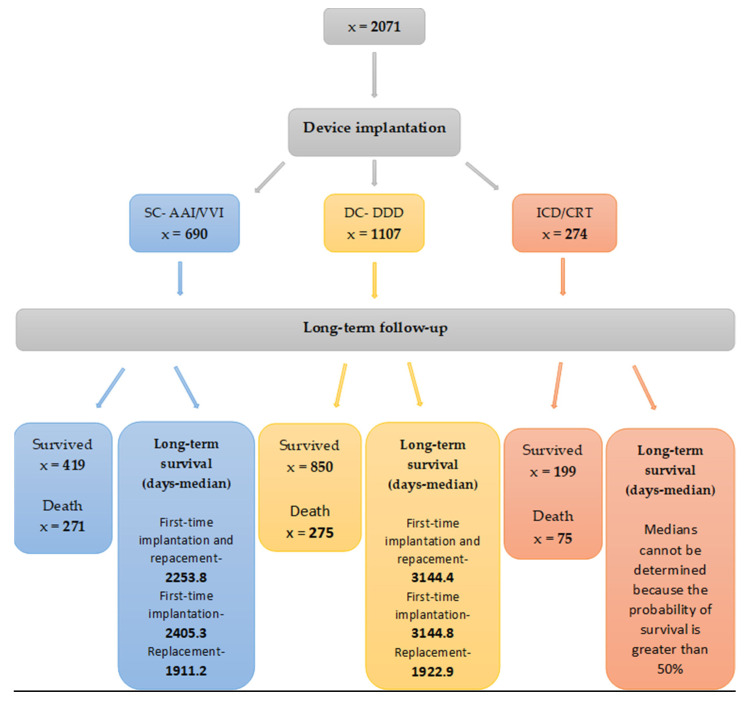
A flowchart of the study with the long-term survival of patients depending on the type of implanted device.

**Figure 2 medicina-57-01357-f002:**
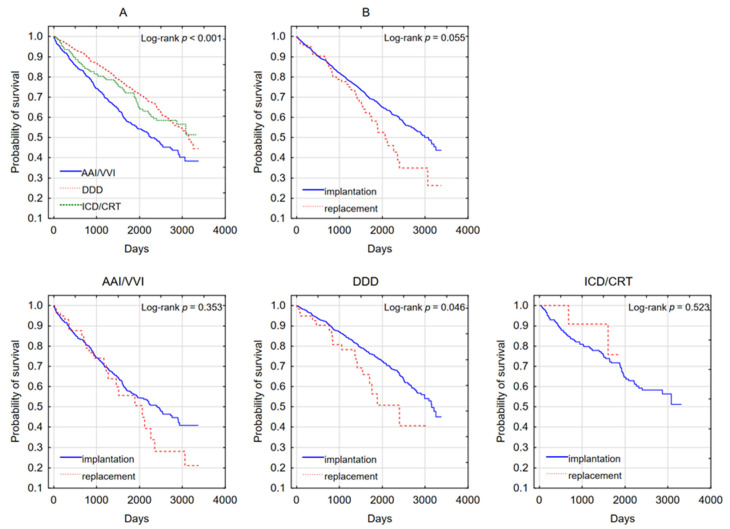
Kaplan–Meier life-table analysis in subgroups depending on the implementation or replacement and type of device. (**A**) The likelihood of patient survival, comparing three types of implanted devices. (**B**) The likelihood of patient survival depending on type of procedure (first implantation or replacement).

**Table 1 medicina-57-01357-t001:** The examined patients’ characteristics, divided by the type of device (implanted data presented as numbers and percentages).

Variable	SC AAI/VVI	DC DDD	ICD/CRT	*p*-Value
Age (years)	80 (74–84)	78 (71–83)	66 (59–74)	<0.001 *
Male	357 51.7%	486 43.9%	234 85.4%	<0.001 *
Heart failure II NYHA class	217 31.5%	238 21.6%	103 37.6%	<0.001 *
Heart failure III NYHA class	119 17.3%	58 5.2%	99 36.1%	<0.001 *
Hypertension	456 66.1%	848 76.6%	140 51.1%	<0.001 *
Diabetes	192 27.8%	310 28.0%	93 33.9%	0.123
Chronic coronary syndrome	193 28.0%	391 35.3%	204 74.5%	<0.001 *
Dilated cardiomyopathy	6 0.9%	5 0.5%	56 20.4%	<0.001 *
Hypertrophic cardiomyopathy	0 0%	4 0.4%	1 0.4%	-
Atrial fibrillation	116 16.8%	350 31.6%	104 38.0%	<0.001 *
History of stroke	94 13.6%	97 8.8%	24 8.8%	0.003 *
Chronic obstructive pulmonary disease	61 8.84%	64 5.78%	36 13.14%	<0.001 *
Chronic kidney disease	116 16.8%	170 15.4%	39 14.2%	0.552
Hyperthyroidism	29 4.2%	38 3.4%	10 3.7%	0.702
Hypothyroidism	32 4.6%	64 5.8%	8 2.9%	0.129
LVEF—primary prevention	-	-	30%	-
LVEF—secondary prevention	-	-	38%	-
Type of procedure	*N*	*N*	*N*	
First-time implantation	630	1049	253	-

* The differences are statistically significant for all comparisons at *p* < 0.05. SC = single chamber; DC = dual chamber; AAI = atrial single-chamber pacemaker; VVI = ventricular single-chamber pacemaker; DDD = dual-chamber pacemaker; ICD = implantable cardioverter defibrillator; CRT = cardiac resynchronisation therapy; NYHA = New York Heart Association; LVEF = left ventricular ejection fraction; AV = atrioventricular; SCD = sudden cardiac death; NS = nonsignificant.

**Table 2 medicina-57-01357-t002:** Primary indications for device implantation (data presented as numbers and percentages).

Primary Indications	SC AAI/VVI	DC DDD	ICD/CRT
Atrial fibrillation with an AV block *	488 (70.7%)	40 (3.6%)	-
AV block III	113 (16.4%)	285 (25.7%)	-
Sick sinus syndrome	63 (9.1%)	520 (47.0%)	-
AV block II t.2	20 (2.9%)	133 (12.0%)	-
AV block 2:1	5 (0.7%)	107 (9.7%)	-
Trifascicular block	1 (0.1%)	1 (0.1%)	-
AV block II t.1	-	19 (1.7%)	-
Alternating bundle branch block	-	2 (0.2%)	-
Cardiac arrest—primary prevention	-	-	205 (74.8%)
Cardiac arrest—secondary prevention	-	-	69 (25.2%)

SC = single chamber; DC = dual chamber; AAI = atrial single-chamber pacemaker; VVI = ventricular single-chamber pacemaker; DDD = dual-chamber pacemaker; ICD = implantable cardioverter defibrillator; CRT = cardiac resynchronisation therapy; AV = atrioventricular. * Atrial fibrillation with an AV block = Bradycardia or inappropriate chronotropic response (due to either an intermittent or complete AV block) associated or reasonably correlated with symptoms. Pacing is indicated in patients with atrial fibrillation and a permanent or paroxysmal third- or high-degree AV block, irrespective of symptoms.

**Table 3 medicina-57-01357-t003:** Multivariate analysis showing the factors affecting survival in three groups: A (SC-AAI/VVI), B (DC-DDD), and C (ICD/CRT).

Analysed Variable	HR	95% CI	*p*-Value
	A (SC-AAI/VVI)		
Chronic kidney disease	1.94	1.45–2.59	<0.001 *
Male	1.62	1.25–2.10	<0.001 *
AV block III	1.59	1.18–2.13	0.002 *
Age (1-year increase)	1.06	1.04–1.08	<0.001 *
Length of hospitalisation	1.05	1.03–1.07	<0.001 *
Stratified by cardiology and urology clinic			
	B (DC-DDD)		
Alternating bundle branch block	39.3	5.03–307.78	<0.001 *
Heart failure III NYHA class	1.78	1.19–2.68	0.005 *
Chronic kidney disease	1.63	1.20–2.22	0.002 *
Diabetes	1.56	1.19–2.05	0.001 *
Male	1.45	1.13–1.86	0.004 *
Age (1-year increase)	1.06	1.04–1.08	<0.001 *
Distance between the place of residence and the implanting centre	0.99	0.99–0.10	0.023 *
Hypertension	0.72	0.55–0.96	0.023 *
Stratified by cardiology clinic			
	C (ICD/CRT)		
Atrial fibrillation with AV block	4.95	1.51–16.25	0.008 *
Hypothyroidism	3.45	1.06–11.26	0.040 *
Age (1-year increase)	1.04	1.02–1.07	<0.001 *
Stratified by cardiology clinic			

* The differences are statistically significant for all comparisons with *p* < 0.05. SC = single-chamber; HR = hazard ratio; CI = confidence interval; DC = dual-chamber; AAI = atrial single-chamber pacemaker; VVI = ventricular single-chamber pacemaker; DDD = dual-chamber pacemaker; ICD = implantable cardioverter defibrillator; CRT = cardiac resynchronisation therapy; NYHA = New York Heart Association; LVEF = left ventricular ejection fraction; AV = atrioventricular; SCD = sudden cardiac death; NS = nonsignificant.

## Data Availability

The data presented in this study are available on reasonable request from the corresponding author.

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
