# Peer review of "Long-Term, Single-Centre Observation of Patients with Cardiac Implantable Electronic Devices"

_medicina, 2021, doi:10.3390/medicina57121357_

Round 1
Reviewer 1 Report
I read this paper with great interest. It is a single-center study aiming to depict the outcomes of patients that had an electrotherapy device implanted, taking into consideration various clinical factors.
A large part of the introduction is a socioeconomic analysis that may affect the prognosis of patients. This does not seem to be a part of the paper's objective and it falls out of scope regarding the paper's goal.
In results, Table 3 is not clear and the results that is trying to communicate are not comprehensive. In what is it expressed, also the categorization should be different.
What does atrial fibrillation with AV block II advanced means?
The results of the multivariate analysis (table 4) are also not comprehensive, it is quite difficult to read the table.
The results are improperly presented and very difficult to read and understand. Also it is quite difficult to understand the association between the results and the primary endpoint of the paper (outcomes/survival of patients).
The discussion part is incoherent.
Although the need for such registries is real, this paper needs a vast review, especially in terms of results presentation and association with the primary endpoint.
I am more than happy to review this paper again in the future.
Author Response
Reviewer I
We wish to express our appreciation for the valuable comments and suggestions for our manuscript entitled “ Long - term single - center observation of patients with cardiac implantable electronic devices”. We have carefully revised the manuscript taking into consideration all the comments. They will contribute to the significant improvement of our manuscript.
We have marked the changes in red. We also used the change tracking feature.
- A large part of the introduction is a socioeconomic analysis that may affect the prognosis of patients. This does not seem to be a part of the paper's objective and it falls out of scope regarding the paper's goal.
Thank you for this comment. We agree with the reviewer’s suggestion. We have removed the socioeconomic analysis from the paper.
- In results, Table 3 is not clear and the results that is trying to communicate are not comprehensive. In what is it expressed, also the categorization should be different.
We thank the reviewer for his comment on the incompleteness of Table 3. The median could not be found in the ICD/CRT group as the probability of survival for the patients was higher than 50%. This table was removed from the paper. In view of the appropriateness of the inclusion of median survival data for each subgroup, and taking into account the reviewer's comment on the inclusion of the study design, Figure 1 presents the study flow chart, which takes into account the numbers of subgroups with the possibility of calculating survival rates. In addition, this figure was enriched with median survival.
- What does atrial fibrillation with AV block II advanced means?
Thank you for this valuable comment. The definition used was not clear. We should have used the diagnosis Atrial Fibrillation with AV block. If slow and regular ventricular rhythm occurs in the course of atrial fibrillation the AV block should be suspected. In cases where there is no persistent AV block or the patient is asymptomatic is impossible to determine the minimal duration of the pause as an indication for stimulation. We have removed the term ‘II advanced’ from the paper and put an explanation to the used term under the table.
- The results of the multivariate analysis (table 4) are also not comprehensive, it is quite difficult to read the table.
Thank you for your comment on Table 4. We agree that the results may have been presented in a vague manner. The goal was to generate multivariate models indicating independent risk factors affecting survival. The models were created separately for each of the 3 subgroups. Therefore, for the time being, we propose to leave this table. For greater clarity, we have specified sections a, b, c in the table and improved the description of the table. We placed the description of results from the multivariable stepwise backward Cox analysis in the chapter Results.
- The results are improperly presented and very difficult to read and understand. Also it is quite difficult to understand the association between the results and the primary endpoint of the paper (outcomes/survival of patients).
Thank you for this comment and earlier comments. We have described the results of the conducted analysis more thoroughly in the chapter Results. We hope that the use of the flow chart and the modifications of the tables have had a positive impact on the readability of the results and at the moment the relationship between the results and the endpoints of the study is sufficiently presented.
- The discussion part is incoherent.
Thank you for the comment. We have modified the discussion in order to improve consistency.
In addition, we have used the professional English Editing service of MDPI to improve the English language. We chose the specialized edition. We hope that this has significantly improved the readability of our manuscript.
Reviewer 2 Report
Can the authors please reduce the length of the introduction ?
Can the authors please add a study flow chart ?
Can the authors please explain how their multivariable analysis was done?
English language needs to improve.
Author Response
Reviewer II
We wish to express our appreciation for the valuable comments and suggestions for our manuscript entitled “ Long - term single - center observation of patients with cardiac implantable electronic devices”. We have carefully revised the manuscript taking into consideration all the comments. They will contribute to the significant improvement of our manuscript.
We have marked the changes in red. We also used the change tracking feature.
- Can the authors please reduce the length of the introduction ?
Thank you for the comment. We agree that the introduction is too long and the socioeconomic paragraphs draw the attention away from the essence of the analysis. We have removed those paragraphs from the paper.
- Can the authors please add a study flow chart ?
Thank you for the valuable comment. Introduction of the study flow chart will improve its clarity. We have placed Figure 1 in the chapter Results which contains the analysis grid together with the results of long-term follow-up.
- Can the authors please explain how their multivariable analysis was done?
Thank you for the comment. In order to obtain the multivariable analysis model, a full model i.e. containing all variables important from the medical point of view was created during the first stage. Next, all statistically irrelevant variables were eliminated. In the next step, the analysis of assumptions of the proportional hazards model was done. Variables that violated the assumption were included into the model as stratifying variables.
For patients with SC-AAI/VVI the Cox multifactorial proportional-hazards model was created, layered because of the variables: cardiology clinic or urology clinic. The model contains the following variables: age of admission, chronic kidney disease, AV block III, gender and length of hospitalization. For patients with DC-DDD the Cox multifactorial proportional-hazards model was created, layered because of the variable cardiology clinic. The model contains the following variables: age of admission, III heart failure NYHA class III, diabetes, chronic kidney disease, alternating bundle branch block, gender, distance between place of living and place of implantation. For patients with ICD/CRT the Cox multifactorial proportional-hazards model was created , layered because of the variable cardiology clinic. The model contains the following variables: age of admission, hypothyroidism and ventricular fibrillation with second degree atrioventricular block.
We have placed results obtained thanks to this mehod in the chapter Results. Table 3 was completed with information concerning stratifying variables.
- English language needs to improve.
Thank you for your comment. We have used the professional English Editing service of MDPI to improve the English language. We chose the specialized edition. We hope that this has significantly improved the readability of our manuscript.
Round 2
Reviewer 1 Report
I read this reviewed version of the paper with great interest. It has greatly improved and I congratulate the authors on that.
I do still have some comments that I would like to address.
- I believe that it needs to be stated clearly both in the Abstract and in the main manuscript that this is a single-centre registry. It is more scientifically bound and states the purpose of the paper in a more clear and straightforward way.
- The authors should include the reason of death for those patients. It should be clear that the reason of death is not medically irrelevant (i.e accident)
- Please explain the headings in Table 3. For example what is HR?
Author Response
Reviewer I
Thank you again for your very valuable comments on our work. We believe that making the suggested changes has significantly improved our text.
- I believe that it needs to be stated clearly both in the Abstract and in the main manuscript that this is a single-centre registry. It is more scientifically bound and states the purpose of the paper in a more clear and straightforward way.
Thank you for this comment. We agree that the information about the single-center nature of the analysis is important. As suggested, we have included this information in the abstract and in the introduction of the paper.
- The authors should include the reason of death for those patients. It should be clear that the reason of death is not medically irrelevant (i.e accident)
Thank you for this comment. Data on patient deaths comes from the State Systems Department of the Ministry of Administration and Digitization. This central registry does not contain information on the causes of patient deaths. On the basis of the document Health status of the Polish population and its determinants, issued by the National Institute of Public Health, it has been assumed that the causes of death among younger people (men aged 25-44 and women aged 10-29) are external, e.g., accidents, suicides and consequences of crimes. In later years, the most common cause of death is cardiovascular disease, especially in men. The population analyzed in our study is the elderly population, which is why we adopted this simplification.
- Please explain the headings in Table 3. For example what is HR?
Thank you for this comment. We have supplemented Table 3 with a description of the abbreviations used.
Kind regards,
Roman Załuska
Reviewer 2 Report
Suggested changes were done by the authors.
Author Response
Reviewer II
Thank you again for your very valuable comments on our work. We believe that making the suggested changes has significantly improved our text.
Kind regards,
Roman Załuska